# C-Arm Cone-Beam CT Virtual Navigation versus Conventional CT Guidance in the Transthoracic Lung Biopsy: A Case-Control Study

**DOI:** 10.3390/diagnostics12010115

**Published:** 2022-01-05

**Authors:** Lian Yang, Yue Wang, Lin Li, Dehan Liu, Xin Wu, Wei Zhang, Feng Pan, Huimin Liang, Chuansheng Zheng

**Affiliations:** 1Department of Radiology, Union Hospital, Tongji Medical College, Huazhong University of Science and Technology, Jiefang Avenue #1277, Wuhan 430022, China; yanglian@hust.edu.cn (L.Y.); m202076089@hust.edu.cn (Y.W.); 2015xh0933@hust.edu.cn (L.L.); 2015xh0934@hust.edu.cn (D.L.); 2009xh1135@hust.edu.cn (X.W.); 2009xh1108@hust.edu.cn (W.Z.); 1988xh0737@hust.edu.cn (H.L.); cszheng@hust.edu.cn (C.Z.); 2Hubei Province Key Laboratory of Molecular Imaging, Jiefang Avenue #1277, Wuhan 430022, China

**Keywords:** lung neoplasms, image-guided biopsy, cone-beam computed tomography, postoperative complications, risk factors

## Abstract

C-arm cone-beam computed tomography (CBCT) virtual navigation-guided lung biopsy has been developed in the last decade as an alternative to conventional CT-guided lung biopsy. This study aims to compare the biopsy accuracy and safety between these two techniques and explores the risk factors of biopsy-related complications. A total of 217 consecutive patients undergoing conventional CT- or C-arm CBCT virtual navigation-guided lung biopsy from 1 June 2018 to 31 December 2019 in this single-center were retrospectively reviewed. Multiple factors (e.g., prior emphysema, lesion size, etc.) were compared between two biopsy techniques. The risk factors of complications were explored by using logistic regression. The patients’ median age and male-to-female ratio were 63 years and 2.1:1, respectively. Eighty-two (82) patients (37.8%) underwent conventional CT-guided biopsies, and the other 135 patients (62.2%) C-arm CBCT virtual navigation-guided biopsies. Compared with patients undergoing C-arm CBCT virtual navigation-guided lung biopsies, patients undergoing conventional CT-guided lung biopsies showed higher needle repositioning rate, longer operation time, and higher effective dose of X-ray (52.4% vs. 6.7%, 25 min vs. 15 min, and 13.4 mSv vs. 7.6 mSv, respectively; *p* < 0.001, each). In total, the accurate biopsy was achieved in 215 of 217 patients (99.1%), without a significant difference between the two biopsy techniques (*p* = 1.000). The overall complication rates, including pneumothorax and pulmonary hemorrhage/hemoptysis, are 26.3% (57/217), with most minor complications (56/57, 98.2%). The needle repositioning was the only independent risk factor of complications with an odds ratio of 6.169 (*p* < 0.001). In conclusion, the C-arm CBCT virtual navigation is better in percutaneous lung biopsy than conventional CT guidance, facilitating needle positioning and reducing radiation exposure. Needle repositioning should be avoided because it brings about more biopsy-related complications.

## 1. Introduction

The conventional CT-guided lung biopsy is one of the pathologically diagnostic methods for solid pulmonary neoplasms, with a diagnostic sensitivity for malignancies of more than 90% [1,2]. It is minimally invasive with excellent safety and can be performed under local anesthesia [2]. It is specifically indicated for peripheral lung lesions that bronchoscopy cannot reach [2,3]. 

Nevertheless, the technique of conventional CT-guided lung biopsy has some shortcomings. First, there is a lack of real-time monitoring during the puncture process [4,5]. The operator can only advance the needle based on the preset route from the initial CT scan [4,5]. If the patient’s respiration is not cooperative, the needle may miss the target lesion during the puncture, and the process must be repeated. As a result, it probably increases the iatrogenic injury to the lung parenchyma, prolongs the procedure time, and enhances the radiation exposure because of numerous CT scans. Second, the needle placement is limited by the axial imaging plane of the CT scanner [6]. Thus, it is difficult to achieve an ideal puncture path setting under conventional CT guidance in some cases. 

Since a decade ago, C-arm cone-beam CT (CBCT) virtual navigation-guided lung biopsy has been developed due to its high technical success rate (>99%), high accuracy (>95%), and good safety [5,7,8,9,10,11,12,13,14]. The C-arm CBCT system consists of flat-panel fluoroscopy and CBCT scanner. It can use 3-dimensional (3D) reconstruction of CBCT imaging to achieve virtual navigation and monitor the needle puncture under fluoroscopy [7,9,10,12,13,14]. The real-time fluoroscopy can give direct feedback of the opacified solid lesion movement due to respiration, facilitating precise respiratory control under the virtual navigation overlay [7,10,14]. Besides this, due to the free 3D rotation of the C-arm, C-arm CBCT can realize free-angle puncture navigation, recognizing the best-preset puncture path [7,8,9,10,11,12,13]. However, there has been a minimal number of studies comprehensively comparing the conventional CT- and C-arm CBCT virtual navigation-guided lung biopsies, especially for the lung lesions firstly indicated for transthoracic biopsies [2,3]. Therefore, our study intends to compare the diagnostic accuracy and safety between these techniques and explore the risk factors of biopsy-related complications. 

## 2. Methods

### 2.1. Patients

The consecutive electronic hospitalization records from 1 June 2018 to 31 December 2019 were retrospectively reviewed in this single center (Western Campus affiliated to Union Hospital, Wuhan, China). A total of 217 patients undergoing conventional CT- or C-arm CBCT virtual navigation-guided lung biopsy for the solid pulmonary lesions were included. The clinical data, including procedure records and pathological results, were collected. 

### 2.2. Pre-Procedure Evaluation

We followed the expert recommendations and international guidelines in lung biopsy evaluation, planning, and performance [2,3]. Before the biopsy, enhanced chest CT and laboratory investigations (e.g., blood routine, coagulation test, etc.) were routinely performed. Further positron emission tomography was carried out if it was hard to differentiate between the neoplasm and post-obstructive atelectasis. After completing the preoperative examinations, a multidisciplinary team (incl. interventionalist, thoracic surgeon, pulmonologist, and anesthetist) evaluated and decided whether the lung biopsy needed to be executed. If lung biopsy was indicated, transthoracic lung biopsy was considered in case of a non-candidate for bronchoscopy, an inaccessible pulmonary lesion by bronchoscopy, a prior non-diagnostic bronchoscopic biopsy, or a subjective requirement by the patient [2]. The choice of CT or CBCT guidance depended on the availability of equipment in the schedule. Before the biopsy, the operators should review the prior enhanced chest CT and other relevant imaging to clarify the target required to be biopsied. In the planning of needle puncture, structures including ribs, vascular vessels, blebs, bullae, central bronchi, and fissures should be dodged [2]. In addition, if a lung lesion demonstrated central cavity or necrosis, the periphery was targeted. The written informed consent from the patient was acquired 24 h before the procedure. All procedures were performed by the same team (HL, FP, LL, and DL) under the supervision of one senior interventionalist (HL with 25 years and one year of experience in CT- and C-arm CBCT virtual navigation-guided biopsy before, respectively). 

### 2.3. Conventional CT Guidance in Lung Biopsy

A 64-slice CT scanner (Philips Ingenuity Core128, Philips Medical Systems, Best, The Netherland) was used. The image acquisition parameters included: 120 kVp with adaptive current modulation, a pitch of 0.999, collimation of 64 × 0.625 mm, a gantry rotation time of 0.5 s, and a DoseRight index of 18. Axial CT images were reconstructed (slice thickness of 1.5 mm and increment of 1.5 mm) with iterative reconstruction (iDose level 5, Philips Healthcare). An initial scan was performed to establish the eligible puncture route to the target lesion under breath-holding after a full aspiration. After locating the target slice, the skin entry point was determined using the cross of a radiopaque grid placed on the patient skin and the perpendicular laser beam (Figure 1a) [2]. After dermal sterilization, draping, and local anesthesia, the patient was asked to hold the breath again, and a 17 G coaxial needle (Bard^®^ TruGuide^®^, 17 G*13.8 cm, Bard Care, Covington, OH, USA) was advanced toward the target following the preset route and depth. Afterward, the chest CT was re-performed to check whether an eligible needle position was reached (Figure 1b). If not, the needle repositioning was carried out until the needle tip reached the target. 

### 2.4. CBCT Guidance in Lung Biopsy

A rotating angiography system (Artis Zee, Siemens Healthcare, Erlangen, Germany) was used to obtain 3D chest CBCT images during the procedure. Similarly, the patient was asked to hold a breath after a full aspiration during the initial image acquisition. A 6 s rotational scan generated 397 projection images with an angular step of 0.5° and a pulse length of 3.2 ms. A tube voltage of 90 kV with a current of 273.5 mA was set up. Then, the projection images were automatically transmitted to a post-processing workstation (Syngo X Workplace, Siemens Healthcare) for multi-planar reconstruction (MPR) of axial, sagittal, and coronal orientations and 3D volume reconstruction. In the next step, the needle path was set up using a commercial plug-in (Syngo iGuide, Siemens Healthcare) [15,16]. After manually selecting the skin entry point and target lesion position in MPR (Figure 2a), a virtual path with displayed angulation and length was generated and overlayed to real-time fluoroscopy to navigate the puncture process. Then, the operator rotated the C-arm to the Bull’s Eye View and turned on the laser navigation system on the flat panel to locate the skin entry point (Figure 2b). After dermal sterilization, draping, and local anesthesia, the patient was asked to hold a breath again. Then, under real-time fluoroscopy monitoring, a 17 G coaxial needle was advanced along the virtual path until the planned target position was reached (Figure 2c). Afterward, a CBCT scan was re-performed to identify whether the needle repositioning was needed (Figure 2d). 

### 2.5. Lung Biopsy

After achieving an ideal position of the coaxial needle, the stylet was removed. Then, a biopsy instrument (Bard^®^ Max-Core^®^ Disposable Core Biopsy Instrument, 18 G*16 cm, Bard Care, Covington, OH, USA) was advanced to obtain samples (so-called “coaxial cutting needle technique”) (Figure 2e) [10,17,18]. Two to eight samples were obtained based on the pathological and other genetic test demands. Afterward, the needle was removed, and the post-biopsy CT or CBCT was routinely acquired to identify biopsy-related complications (Figure 1c and Figure 2f). 

### 2.6. Study Goals

We compared prior emphysema history, lesion size, technical success, needle repositioning, operation time, an effective X-ray dose, complications, and biopsy accuracy between patients undergoing conventional CT- and C-arm CBCT virtual navigation-guided lung biopsies. The lesion size was defined as an average of long- and short-axis measurements in the maximum lesion section of axial CT images following the radiological statement from the Fleischner Society (Figure 3a) [19]. Technical success was defined as satisfactory biopsy materials that qualified for pathologic analysis [20]. The needle repositioning was defined as the pull-back adjustment or re-puncture when the needle tip did not reach the planning target [5,6]. During the procedure, operation time was recorded from the start of the initial CT/CBCT scan to the end of the post-biopsy CT/CBCT scan. The effective dose of conventional CT was calculated by using a dose-linear product (DLP)*κ-factor and fluoroscopy/CBCT by using a dose-area product (DAP)*dose conversion coefficient [21]. For conventional chest CT, κ-factor was 0.0146 mSv/mGy*cm [22]. For chest fluoroscopy/CBCT of the Artis system, the dose conversion coefficient was 0.0017 mSv/μGy*m^2^ [10,14]. The accurate biopsy was defined as the biopsy pathological result that was: 1. confirmed by the following surgery; or 2. supported by subsequent clinical course for at least one year (e.g., growth or metastasis of malignancies, stable or regression of benign lesions, etc.) [6,7,10,23]. 

The biopsy-related complications were collected and classified from *Grade-A* to -*E* under the standards of the Society of Interventional Radiology [24]. A minor complication was defined as *Grade-A* or *-B* complication, and a major complication as *Grade-C* to *-E* complication [24]. Besides this, the risk factors of complication were comprehensively explored, including prior emphysema history, patient position during the biopsy, lesion size, needle repositioning, lesion-skin distance along the needle path (from lesion border to skin) (Figure 3b), lesion-pleural distance along the needle path (from lesion border to pleura) (Figure 3c), needle-pleural angle (Figure 3d), and the number of obtained samples [4,5,19,20,23]. Two independent radiologists performed all measurements (LY and YW, with 23- and 2-year experience in thoracic radiology, respectively), and the average of the measurements was involved in the final analysis. 

### 2.7. Statistical Analysis

Statistical analyses were performed using SPSS Statistics Software (version 26; IBM, New York, NY, USA). Quantitative data were presented as median with inter-quantile range (IQR), while the counting data were presented as count with the percentage of the total. The quantitative data comparisons were evaluated using Mann–Whitney tests, according to the non-normal distribution assessed by Shapiro–Wilk tests. Chi-square tests were performed to assess categorical variables between different groups, and Fisher’s exact test was implemented instead of the Chi-square test if the expected count was less than five. Univariate and multivariate logistic regression tests using forward conditional methods were performed to investigate the independent risk factors for biopsy-related complications. Odds ratio (OR) with 95% confidence interval (95%CI) was calculated. Inter-observer agreement of the measurements was assessed using intraclass correlation efficient (ICC) analysis by applying a two-way random model. All tests were two-sided, and a *p*-value of <0.01 was defined as a statistical significance. 

## 3. Results

### 3.1. Patient Characteristics

The patients’ median age and male-to-female ratio were 63 years (IQR: 54–69 years) and 2.1:1, respectively. More than one-fifth of patients (60/217, 27.6%) had prior emphysema history. Among them, 82 patients (82/217, 37.8%) underwent conventional CT-guided lung biopsies, while the other 135 patients (135/217, 62.2%) underwent C-arm CBCT virtual navigation-guided lung biopsies. The biopsied lesions were mainly located in the right and left upper lung lobes (31.8% and 28.6%, respectively). The median lesion size was 40.0 mm (IQR: 26.8–54.8 mm). More details are summarized in Table 1.

### 3.2. Comparisons between Conventional CT- and C-arm CBCT Virtual Navigation-Guided Lung Biopsies

The technical success of the biopsy was achieved in all patients (217/217, 100.0%). Compared with patients undergoing C-arm CBCT virtual navigation-guided lung biopsies, patients undergoing conventional CT-guided lung biopsies showed higher needle repositioning incidence, longer operation time, a higher effective dose of X-ray, and less obtained samples (52.4% vs. 6.7%, 25 min vs. 15 min, 13.4 mSv vs. 7.6 mSv, and four vs. 5, respectively; *p* < 0.001, each) (Table 2). On the other hand, our results showed no significant differences in age, sex, lesion size, lesion-skin/lesion-pleural distance along the needle path, needle-pleural angle, complications, and biopsy accuracy between the two biopsy groups (Table 2). Overall, the accurate biopsy was achieved in 215 of 217 patients (99.1%) (Table 2). Two (2) patients (2/217, 0.9%) diagnosed with inflammation by biopsies were finally confirmed with malignancies after surgical resection (one was small cell carcinoma; the other was the metastasis of clear cell renal cell carcinoma) (Table 2). 

### 3.3. Risk Factor of Complications

The rates of pneumothorax, pulmonary hemorrhage/hemoptysis, and overall complications were 14.3% (31/217), 21.7% (47/217), and 26.3% (57/217), respectively. The minor complications occupied 98.2% (56/57) of the overall complications (Table 2). Multiple factors were compared between patients with and without biopsy-related complications (incl. pneumothorax and pulmonary hemorrhage/hemoptysis) (Table 3). As a result, the incidence of biopsy-related complications was significantly higher in patients with intraprocedural needle repositioning than in those without repositioning (55.8% vs. 17.0%, *p* < 0.001) (Table 3). Further univariate and multivariate logistic regression analyses revealed needle repositioning was the only independent risk factor of biopsy-related complications; the ORs of overall complications, pneumothorax, and pulmonary hemorrhage/hemoptysis were 6.169, 10.463, and 6.857, respectively (*p* < 0.001, each) (Table 4). 

## 4. Discussion

Our retrospective study finds that the C-arm CBCT virtual navigation with real-time fluoroscopy monitoring in lung biopsy could reduce the incidence of needle repositioning compared to conventional CT guidance (6.7% vs. 52.4%, *p* < 0.001). It shortened the operation time and lowered the effective dose of X-ray (15 min vs. 25 min and 7.6 mSv vs. 13.4 mSv, respectively; *p* < 0.001, both). The technical success rates were 100% in both biopsies, with biopsy accuracies of over 98%. The overall complication incidence was 26.3% (57/217), most of which are minor complications (56/57, 98.2%). Further logistic regression analysis revealed that needle repositioning during the procedure was the only independent risk factor for biopsy-related complications (OR: 6.169, *p* < 0.001). Our results suggest that needle repositioning increases the risk of pneumothorax and pulmonary hemorrhage/hemoptysis by approximately 10 and 7 times, respectively. 

In a previous study involving a total of 58 patients, C-arm CBCT virtual navigation presented more favorable merits than conventional CT guidance in biopsies at different organs [6]. It brought about less needle repositioning, similar to the results in lung biopsies of our study [6]. However, the number of needle repositioning and operation time in our conventional CT-guidance group was significantly lower and shorter than in the previous report (One vs. 2 and 25 min vs. >30 min, respectively), indicating the importance of proficient skills for biopsies under conventional CT guidance [6]. Even so, C-arm CBCT virtual navigation in our study still significantly reduced the needle repositioning incidence compared with conventional CT guidance (6.7% vs. 52.4%, *p* < 0.001). These findings suggest C-arm CBCT virtual navigation is better than conventional CT guidance in coaxial needle positioning. Although direct comparison between the effective dose of conventional CT and CBCT is impossible, standard algorithmic conversion was established from previous studies [10,14,21,22]. As a result, a median effective dose was 7.6 mSv under C-arm CBCT virtual navigation with a 43.3% reduction compared with conventional CT, similar to previous reports [6,10,12,14,25,26]. As an alternative, fluoroscopic CT guidance can also monitor the needle positioning in real-time [25,26,27,28]. However, fluoroscopic CT delivers higher radiation doses to both the patient and operator than conventional CT, much more than CBCT [18,29]. So, we do not prefer it in our center. 

Although controlled studies are few so far, by comparing with other research, it was reported that the biopsy accuracy was higher in C-arm CBCT virtual navigation than conventional CT guidance (98.2% vs. 83.7%) when diagnosing small lung solid lesions less than 15–20 mm [4,10]. However, our study did not find a significant difference in the biopsy accuracy between these two techniques (>98%, both), similar to one previous study [26]. A larger lesion size (median: 40 mm) with a shorter lesion-pleural distance (median: 14 mm) in our cohort probably accounted for this discrepancy from previous studies, because it was reported that the benefit of increased biopsy accuracy under C-arm CBCT virtual navigation was gradually demolished with the increase of the lesion size [4,9,10,11]. After all, the small lesion size is a risk factor of technical failure and false-negative results in either CT- or CBCT-guided lung biopsy [4,10]. 

In accordance with previous studies, we identified pneumothorax, pulmonary hemorrhage, and hemoptysis were the most common biopsy-related complications with an average incidence of about 25% [2,3,7,10,11,12,14,20,30]. A major complication rate of only 2.4–3.8% was reported, indicating most complications did not require additional treatment [7,10]. However, there was a significant variation of the reported complication incidence, such as the pneumothorax rate of 0–61% and the pulmonary hemorrhage/hemoptysis rate of 0–41% [2,3,7,10,11,12,14,20,30]. Nevertheless, unlike published studies, pulmonary hemorrhage/hemoptysis had a higher incidence than pneumothorax (21.7% vs. 14.3%) in our cohort [10,12,14,20]. It probably ascribes to: 1. more samples were obtained in our cohort than in previous studies (5 vs. 2–3); and 2. blebs or bullae were avoided in the puncture process [10,12,14,20]. Additionally, we did not observe other complications, including air embolism, hemothorax, and tumoral seeding [2]. 

Our study revealed that needle repositioning significantly increased the risk of pneumothorax and pulmonary hemorrhage/hemoptysis, likely because it resulted in multiple transverses of the lung parenchyma and pleura [6]. Thus, reducing needle manipulations is essential in a biopsy, no matter which guidance is used. Furthermore, extended lesion-pleura distance was also documented to be significantly associated with the increased risks of pneumothorax and pulmonary hemorrhage/hemoptysis [2,4,14,31]. However, our study does not support this finding, probably because bronchoscopic biopsy is more recommended to obtain samples for central lung lesions, resulting in the majority of the peripheral lung lesions in our cohort [2]. It can also explain why rare major complication (1/217, 0.5%) was observed in our cohort. In addition, some other risk factors were once reported. For instance, ground-glass nodule (GGN) was associated with pulmonary hemorrhage/hemoptysis and emphysema with pneumothorax [2,10,14,20]. However, GGN was not biopsied in our center following the guidelines [3]. Moreover, to reduce the biopsy-related complications, methods such as positioning a biopsy side down and puncture under an acute needle-pleural angle (<51°) were suggested [20,23]. However, our data did not support these findings because a rare lateral recumbent position and larger needle-pleural angle (median: 67.5°) were applied. 

Our study has several limitations. First, it was retrospectively carried out in a single center, and all patients underwent biopsies by the same interventionalists team. However, our results indicate that even if the operators are more skillful in conventional-CT guided biopsy, C-arm CBCT virtual navigation can still facilitate the accurate needle puncture, reducing the need for needle repositioning. Second, small or central lung lesions were rarely involved in our cohort, bringing about some discrepancies from previous studies [4,10]. In these cases, surgical resection or bronchoscopic biopsy was more recommended following guidelines or recommendations [2,3,32]. Third, histologic confirmation from surgical resection did not apply to all patients, similar to most previous studies [6,7,10,23]. In this case, a long-term follow-up (at least one year) was necessary to identify the malignant or benign diagnosis [6,7,10,23]. 

## 5. Conclusions

This retrospective case-control study demonstrates that both the conventional CT- and C-arm CBCT virtual navigation-guided lung biopsies are safe and accurate. However, the C-arm CBCT virtual navigation facilitates the needle positioning and reduces the radiation dose, compared to conventional CT guidance. Since needle repositioning significantly increased the incidence of biopsy-related complications, as revealed in our study, C-arm CBCT virtual navigation can be more suggested in transthoracic lung biopsy to reduce the necessity of needle repositioning and the subsequent complications. 

## Figures and Tables

**Figure 1 diagnostics-12-00115-f001:**
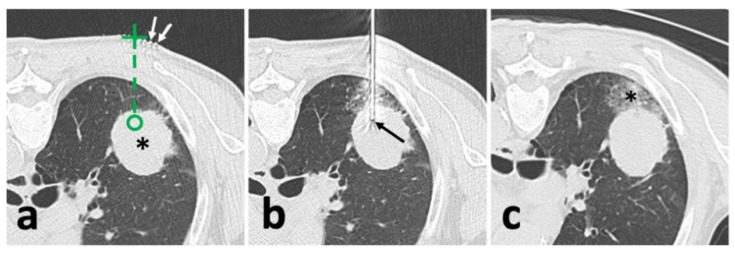
Illustration of conventional CT-guided biopsy. (**a**), localization with opaque grid marking (white arrow) and needle route planning (green dash line) targeting the solid lesion (*); (**b**), after needle puncture, CT showed an appropriate location of the needle tip (black arrow) within the solid lesion; (**c**), after removal of the needle and biopsy instrument, post-biopsy CT demonstrated pulmonary hemorrhage (*) alongside the puncture route without the findings of pneumothorax.

**Figure 2 diagnostics-12-00115-f002:**
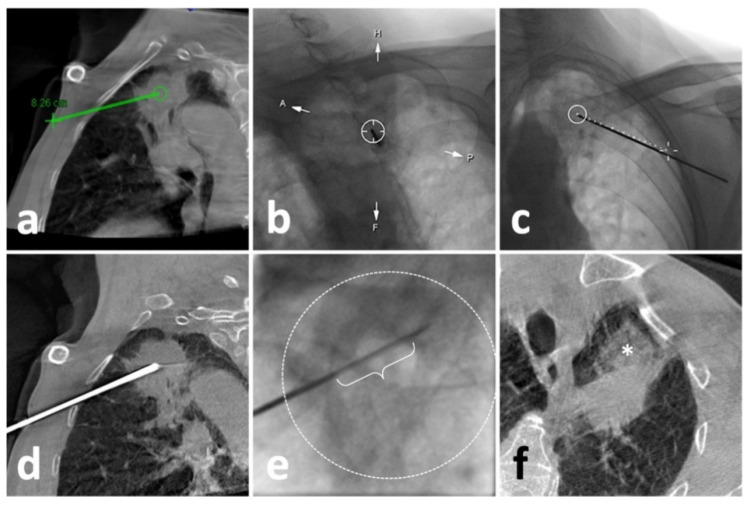
Illustration of C-arm CBCT virtual navigation-guided biopsy. (**a**), localization of the lesion (solid mass at left upper lobe) and needle route planning (green line); (**b**), fluoroscopy-guided puncture under virtual navigation at Bull’s Eye View Position; the middle circle indicates the target which was accurately punctured by a 17-G needle; (**c**), Fluoroscopic check of the needle, which matched perfectly with the virtual path (dash line); (**d**), a re-examination of CBCT showed eligible needle tip within the solid lesion; (**e**), the performance of biopsy under real-time fluoroscopy, in which the sampling part of the biopsy instrument (curly bracket) showed appropriate location within the opacified lesion (dash circle); (**f**), the post-biopsy CBCT demonstrated pulmonary hemorrhage (*) alongside the puncture route without the findings of pneumothorax.

**Figure 3 diagnostics-12-00115-f003:**
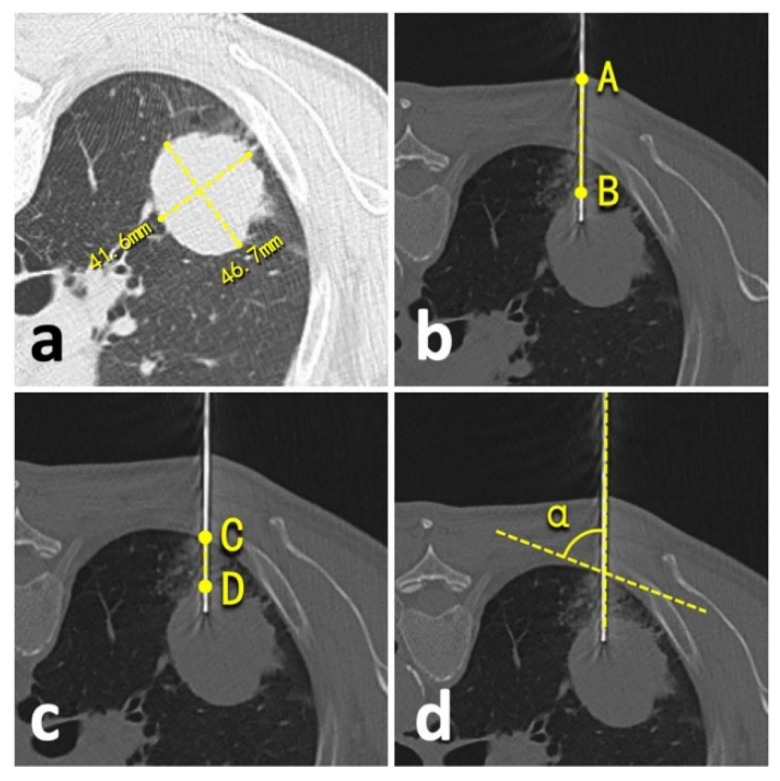
Illustration of quantitative measurements. (**a**), lesion size; (**b**), lesion-skin distance; (**c**), lesion-pleural distance; (**d**), needle-pleural angle.

**Table 1 diagnostics-12-00115-t001:** Basic characteristics.

	Results, n = 217.
Sex	
Male	146 (67.3%)
Female	71 (32.7%)
Age (years) (IQR)	63 (54–69)
Prior emphysema history	60 (27.6%)
Pleural effusion	2 (0.9%)
Lung lesion location ^1^	
Left upper lobe	62 (28.6%)
*Apicoposterior segment (B1/2)*	*30 (13.8%)*
*Anterior segment (B3)*	*13 (6.0%)*
*Superior lingular segment (B4)*	*11 (5.1%)*
*Inferior lingular segment (B5)*	*8 (3.7%)*
Left lower lobe	32 (14.7%)
*Superior segment (B6)*	*8 (3.7%)*
*Anteromedial segment (B7/8)*	*9 (4.1%)*
*Lateral segment (B9)*	*6 (2.8%)*
*Posterior segment (B10)*	*9 (4.1%)*
Right upper lobe	69 (31.8%)
*Apical segment (B1)*	*32 (14.7%)*
*Posterior segment (B2)*	*24 (11.1%)*
*Anterior segment (B3)*	*13 (6.0%)*
Right middle lobe	10 (4.6%)
*Lateral segment (B4)*	*8 (3.7%)*
*Medial segment (B5)*	*2 (0.9%)*
Right lower lobe	44 (20.3%)
*Superior segment (B6)*	*13 (6.0%)*
*Medial segment (B7)*	*3 (1.4%)*
*Anterior segment (B8)*	*11 (5.1%)*
*Lateral segment (B9)*	*8 (3.7%)*
*Posterior segment (B10)*	*9 (4.1%)*
Lesion size (mm) ^2^	40.0 (26.8–54.8)
Biopsy techniques	
Conventional CT-guided lung biopsy	82 (37.8%)
C-arm CBCT virtual navigation-guided lung biopsy	135 (62.2%)

Note: Quantitative data were presented as median with inter-quantile range (IQR) and the counting data were presented as count with the percentage of the total in parenthesis. ^1^ The lesion location was defined as the segment of the lesion center. ^2^ The lesion size was defined as an average of long- and short-axis measurements in the maximum lesion section of the axial CT image.

**Table 2 diagnostics-12-00115-t002:** Comparisons between conventional CT- and C-arm CBCT virtual navigation-guided lung biopsies.

	Total, n = 217.	Conventional CT-Guided Lung Biopsy, n = 82.	C-Arm CBCT Virtual Navigation-Guided Lung Biopsy, n = 135.	*p*-Value ^1^
Age (years)	63 (54–69)	62 (54–70)	63 (53–68)	1.000
≤63	115 (53.0%)	47 (57.3%)	68 (50.4%)	0.320
>63	102 (47.0%)	35 (42.7%)	67 (49.6%)	
Sex				
Male	146 (67.3%)	53 (64.6%)	93 (68.9%)	0.517
Female	71 (32.7%)	29 (35.4%)	42 (31.1%)	
Prior emphysema history	60 (27.6%)	16 (19.5%)	44 (32.6%)	0.037
Lesion size (mm) ^2,3^	40.0 (26.8–54.8) ^4^	42.3 (30.3–54.8)	36.8 (23.8–54.8)	0.250
≤40.0	110 (50.7%)	36 (43.9%)	74 (54.8%)	0.119
>40.0	107 (49.3%)	46 (56.1%)	61 (45.2%)	
Technical success	217 (100.0%)	82 (100.0%)	135 (100.0%)	1.000
Position				
Supine	113 (52.1%)	39 (47.6%)	74 (54.8%)	0.396
Prone	99 (45.6%)	40 (48.8%)	59 (43.7%)	
Lateral recumbent	5 (2.3%)	3 (3.7%)	2 (1.5%)	
Number of needle repositioning	0 (0–0)	1 (0–2)	0 (0–0)	<0.001
Needle repositioning	52 (24.0%)	43 (52.4%)	9 (6.7%)	<0.001
Lesion-skin distance along the needle path (mm) ^3^	51.7 (37.9–67.5) ^5^	51.3 (38.2–64.5)	51.8 (35.1–68.4)	0.865
≤51.7	108 (49.8%)	41 (50.0%)	67 (49.6%)	0.958
>51.7	109 (50.2%)	41 (50.0%)	68 (50.4%)	
Lesion-pleural distance along the needle path (mm) ^3^	14.4 (0.0–28.7) ^6^	14.1 (0.0–31.3)	15.2 (0.0–27.1)	0.349
=0.0	75 (34.6%)	24 (29.3%)	51 (37.8%)	0.438
>0.0 and ≤22.7	70 (32.3%)	29 (35.4%)	41 (30.4%)	
>22.7	72 (33.2%)	29 (35.4%)	43 (31.9%)	
Needle-pleural angle (°) ^3^	67.5 (55.0–80.1) ^7^	65.0 (51.5–77.4)	69.4 (55.9–81.2)	0.198
≤67.5	109 (50.2%)	45 (54.9%)	64 (47.4%)	0.286
>67.5	108 (49.8%)	37 (45.1%)	71 (52.6%)	
Number of obtained samples	5 (4–5)	4 (4–5)	5 (5–6)	<0.001
≤5	173 (79.7%)	80 (97.6%)	93 (68.9%)	<0.001
>5	44 (20.3%)	2 (2.4%)	42 (31.1%)	
Operation time (min)	18 (14–25)	25 (21–31)	15 (12–20)	<0.001
Effective dose of X-ray (mSv)	8.9 (7.1–12.7)	13.4 (10.9–17.4)	7.6 (6.0–9.5)	<0.001
Complications	57 (26.3%)	27 (32.9%)	30 (22.2%)	0.082
Pneumothorax	31 (14.3%)	15 (18.3%)	16 (11.9%)	0.189
Pulmonary hemorrhage/hemoptysis	47 (21.7%)	25 (30.5%)	22 (16.3%)	0.014
Severity grades of complication ^8^				
Minor complications	56 (25.8%)	26 (31.7%)	30 (22.2%)	0.121
*A*	25 (11.5%)	9 (11.0%)	16 (11.9%)	0.098
*B*	31 (14.3%)	17 (20.7%)	14 (10.4%)	
Major complications	1 (0.5%)	1 (1.2%)	0 (0.0%)	0.121
*C*	1 (0.5%)	1 (1.2%)	0 (0.0%)	0.098
Pathological results of biopsies				
Adenocarcinoma	98 (45.2%)	38 (46.3%)	60 (44.4%)	0.714
Non-specific chronic inflammation	34 (15.7%)	16 (19.5%)	18 (13.3%)	
Squamous cell carcinoma	29 (13.4%)	11 (13.4%)	18 (13.3%)	
Tuberculosis	15 (6.9%)	5 (6.1%)	10 (7.4%)	
Malignant cell type not specified	11 (5.1%)	4 (4.9%)	7 (5.2%)	
Adenosquamous carcinoma	10 (4.6%)	3 (3.7%)	7 (5.2%)	
Small cell carcinoma	9 (4.1%)	3 (3.7%)	6 (4.4%)	
Organizing pneumonia	5 (2.3%)	0 (0.0%)	5 (3.7%)	
Fungal infection	4 (1.8%)	2 (2.4%)	2 (1.5%)	
Pneumoconiosis	2 (0.9%)	0 (0.0%)	2 (1.5%)	
Accurate biopsy ^9^	215 (99.1%) ^10^	81 (98.8%)	134 (99.3%)	1.000

Note: Quantitative data were presented as median with inter-quantile range (IQR) and the counting data were presented as count with the percentage of the total in parenthesis. One or two cutpoints of quantitative variables were set up based on equal percentiles. ^1^
*p*-values comparing conventional CT- and C-arm CBCT virtual navigation-guided biopsies were determined using the Mann–Whitney test, Chi-square test, or Fisher’s exact test. ^2^ The lesion size was defined as an average of long- and short-axis measurements in the maximum lesion section of the axial CT image. ^3^ Average of the measurements by two radiologists were involved in the statistical analysis. ^4^ ICC was 0.974 (95%CI: 0.964–0.981, *p* < 0.001). ^5^ ICC was 0.983 (95%CI: 0.978–0.987, *p* < 0.001). ^6^ ICC was 0.973 (95%CI: 0.965–0.980, *p* < 0.001). ^7^ ICC was 0.956 (95%CI: 0.943–0.966, *p* < 0.001). ^8^ The biopsy-related complications were classified under the updated standards of the Society of Interventional Radiology: *A*-No therapy, no sequences; *B*-Nominal therapy, no consequence, includes overnight admission for observation only; and *C*-Requires therapy, minor hospitalization (<48 h); in which minor complication was defined as *Grade-A* or *-B* complication, and major complication as *Grade-C* complication. ^9^ Accurate biopsy was defined as following surgical pathological diagnosis or subsequent clinical course for at least one year (e.g., growth or metastasis of malignant lesion, stable or regression of benign lesion) that were consistent with the biopsy pathological results. ^10^ Two patients diagnosed with non-specific chronic inflammation by biopsies were finally confirmed with malignancies by surgical pathology (one was small cell carcinoma; the other was the metastasis of clear cell renal cell carcinoma).

**Table 3 diagnostics-12-00115-t003:** Comparisons of factors between patients with and without biopsy-related complications.

	Overall Complications		*p*-Value ^1^	Pneumothorax		*p*-Value ^1^	Pulmonary Hemorrhage/Hemoptysis		*p*-Value ^1^
	None, 160/217 (73.7%)	Yes, 57/217 (26.3%)		None, 186/217 (85.7%)	Yes, 31/217 (14.3%)		None, 170/217 (78.3%)	Yes, n = 47/217 (21.7%)	
Age (years)	62 (53–69)	64 (55–69)	0.175	62 (53–69)	64 (61–67)	0.075	62 (53–68)	64 (55–71)	0.096
≤63	89/115 (77.4%)	26/115 (22.6%)	1.193	104/115 (90.4%)	11/115 (9.6%)	0.035	94/115 (81.7%)	21/115 (18.3%)	0.197
>63	71/102 (69.6%)	31/102 (30.4%)		82/102 (80.4%)	20/102 (19.6%)		76/102 (74.5%)	26/102 (25.5%)	
Prior emphysema history									
None	118/157 (75.2%)	39/157 (24.8%)	0.440	135/157 (86.0%)	22/157 (14.0%)	0.853	124/157 (79.0%)	33/157 (21.0%)	0.711
Yes	42/60 (70.0%)	18/60 (30.0%)		51/60 (85.0%)	9/60 (15.0%)		46/60 (76.7%)	14/60 (23.3%)	
Guidance									
Conventional CT	55/82 (67.1%)	27/82 (32.9%)	0.082	67/82 (81.7%)	15/82 (18.3%)	0.189	57/82 (69.5%)	25/82 (30.5%)	0.014
CBCT	105/135 (77.8%)	30/135 (22.2%)		119/135 (88.1%)	16/135 (11.9%)		113/135 (83.7%)	22/135 (16.3%)	
Lesion size (mm) ^2,3^	39.9 (27.3–54.5)	40.0 (26.5–54.8)	0.755	40.1 (26.8–54.8)	37.0 (24.5–55.3)	0.696	40.1 (27.8–55.3)	39.0 (24.5–53.8)	0.462
≤40.0	81/110 (73.6%)	29/110 (26.4%)	0.974	93/110 (84.5%)	17/110 (15.5%)	0.618	85/110 (77.3%)	25/110 (22.7%)	0.698
>40.0	79/107 (73.8%)	28/107 (26.2%)		93/107 (86.9%)	14/107 (13.1%)		85/107 (79.4%)	22/107 (20.6%)	
Position									
Supine	85/113 (75.2%)	28/113 (24.8%)	0.803	100/113 (88.5%)	13/113 (11.5%)	0.468	89/113 (78.8%)	24/113 (21.2%)	0.981
Prone	71/99 (71.7%)	28/99 (28.3%)		82/99 (82.8%)	17/99 (17.2%)		77/99 (77.8%)	22/99 (22.2%)	
Lateral recumbent	4/5 (80.0%)	1/5 (20.0%)		4/5 (80.0%)	1/5 (20.0%)		4/5 (80.0%)	1/5 (20.0%)	
Number of needle repositioning	0 (0–0)	1 (0–1)	<0.001	0 (0–0)	1 (0–2)	<0.001	0 (0–0)	1 (0–2)	<0.001
Needle repositioning									
None	137/165 (83.0%)	28/165 (17.0%)	<0.001	154/165 (93.3%)	11/165 (6.7%)	<0.001	144/165 (87.3%)	21/165 (12.7%)	<0.001
Yes	23/52 (44.2%)	29/52 (55.8%)		32/52 (61.5%)	20/52 (38.5%)		26/52 (50.0%)	26/52 (50.0%)	
Lesion-skin distance along the needle path (mm) ^3^	50.4 (34.6–67.5)	55.5 (41.5–67.5)	0.214	50.8 (35.0–66.4)	56.9 (47.0–70.9)	0.053	51.4 (35.1–68.2)	53.4 (39.6–65.9)	0.556
≤51.7	84/108 (77.8%)	24/108 (22.2%)	0.178	97/108 (89.8%)	11/108 (10.2%)	0.086	86/108 (79.6%)	22/108 (20.4%)	0.646
>51.7	76/109 (69.7%)	33/109 (30.3%)		89/109 (81.7%)	20/109 (18.3%)		84/109 (77.1%)	25/109 (22.9%)	
Lesion-pleural distance along the needle path (mm) ^3^	13.6 (0.0–27.3)	16.3 (0.0–32.4)	0.228	13.1 (0.0–27.5)	19.5 (6.4–38.2)	0.046	14.3 (0.0–28.1)	14.5 (0.0–31.1)	0.550
=0.0	60/75 (80.0%)	15/75 (20.0%)	0.277	69/75 (92.0%)	6/75 (8.0%)	0.149	62/75 (82.7%)	13/75 (17.3%)	0.351
>0.0 and ≤22.7	48/70 (68.6%)	22/70 (31.4%)		57/70 (81.4%)	13/70 (18.6%)		51/70 (72.9%)	19/70 (27.1%)	
>22.7	52/72 (72.2%)	20/72 (27.8%)		60/72 (83.3%)	12/72 (16.7%)		57/72 (79.2%)	15/72 (20.8%)	
Needle-pleural angle (°) ^3^	68.7 (55.5–80.7)	65.0 (50.3–77.2)	0.298	67.9 (55.0–80.2)	63.4 (51.0–79.1)	0.576	68.7 (55.9–80.4)	63.7 (50.3–77.0)	0.262
≤67.5	76/109 (69.7%)	33/109 (30.3%)	0.178	92/109 (84.4%)	17/109 (15.6%)	0.579	81/109 (74.3%)	28/109 (25.7%)	0.148
>67.5	84/108 (77.8%)	24/108 (22.2%)		94/108 (87.0%)	14/108 (13.0%)		89/108 (82.4%)	19/108 (17.6%)	
Number of obtained samples	5 (4–5)	5 (4–5)	0.605	5 (4–5)	5 (4–5)	0.616	5 (4–5)	5 (4–5)	0.287
≤5	126/173 (72.8%)	47/173 (27.2%)	0.550	148/173 (85.5%)	25/173 (14.5%)	0.890	132/173 (76.3%)	41/173 (23.7%)	0.148
>5	34/44 (77.3%)	10/44 (22.7%)		38/44 (86.4%)	6/44 (13.6%)		38/44 (86.4%)	6/44 (13.6%)	

Note: Quantitative data were presented as median with inter-quantile range (IQR) and the counting data were presented as count/row total with the percentage of the row total in parenthesis. One or two cutpoints of quantitative variables were set up based on equal percentiles. ^1^
*p* values comparing patients with and without complications were determined with Mann–Whitney test or Chi-square test. ^2^ The lesion size was defined as an average of long- and short-axis measurements in the maximum lesion section of the axial CT image. ^3^ Average of the measurements by two radiologists were involved in the statistical analysis.

**Table 4 diagnostics-12-00115-t004:** Univariate and multivariate logistic regression analysis.

	Overall Complications		Pneumothorax		Pulmonary Hemorrhage/Hemoptysis	
	OR (95% CI)	*p*-Value ^1^	OR (95% CI)	*p*-Value ^1^	OR (95% CI)	*p*-Value ^1^
Univariate logistic regression analysis						
Age (years)						
>63 vs. ≤63	1.495 (0.814–2.744)	0.195	2.306 (1.046–5.084)	0.038	1.531 (0.800–2.932)	0.199
Prior emphysema history						
Yes vs. None	1.297 (0.670–2.509)	0.441	1.083 (0.468–2.508)	0.853	1.144 (0.562–2.328)	0.711
Guidance						
Conventional CT vs. CBCT	1.718 (0.930–3.174)	0.084	1.665 (0.775–3.580)	0.192	2.253 (1.170–4.339)	0.015
Lesion size (mm)						
≤40.0 vs. >40.0	1.010 (0.552–1.849)	0.974	1.214 (0.566–2.606)	0.618	1.136 (0.595–2.170)	0.699
Position						
Prone vs. Supine	1.197 (0.650–2.206)	0.564	1.595 (0.732–3.475)	0.240	1.060 (0.551–2.038)	0.862
Lateral recumbent vs. Supine	0.759 (0.081–7.076)	0.809	1.923 (0.199–18.544)	0.572	0.927 (0.099–8.684)	0.947
Number of needle repositioning						
+1	1.686 (1.315–2.163)	<0.001	1.480 (1.189–1.844)	<0.001	1.713 (1.341–2.188)	<0.001
Needle repositioning						
Yes vs. None	6.169 (3.120–12.198)	<0.001	8.750 (3.821–20.035)	<0.001	6.857 (3.369–13.957)	<0.001
Lesion-skin distance along the needle path (mm) ^3^						
>51.7 vs. ≤51.7	1.520 (0.825–2.798)	0.179	1.982 (0.899–4.366)	0.090	1.163 (0.609–2.222)	0.647
Lesion-pleural distance along the needle path (mm) ^3^						
>0.0 and ≤22.7 vs. =0.0	1.833 (0.859–3.913)	0.117	2.623 (0.937–7.339)	0.066	1.777 (0.801–3.942)	0.157
>22.7 vs. =0.0	1.538 (0.716–3.308)	0.270	2.300 (0.814–6.502)	0.116	1.255 (0.550–2.864)	0.589
Needle-pleural angle (°) ^3^						
>67.5 and ≤67.5	0.658 (0.357–1.212)	0.179	0.806 (0.376–1.730)	0.580	0.618 (0.321–1.190)	0.150
Number of obtained samples						
>5 vs. ≤5	0.788 (0.361–1.721)	0.551	0.935 (0.358–2.440)	0.890	0.508 (0.201–1.288)	0.154
Multivariate logistic regression analysis ^2^						
Age (years)						
>63 vs. ≤63			3.187 (1.310–7.752)	0.011		
Needle repositioning						
Yes vs. None	6.169 (3.120–12.198)	<0.001	10.463 (4.363–25.090)	<0.001	6.857 (3.369–13.957)	<0.001

Note: OR (95% CI) indicated an odds ratio with a 95% confidence interval in parenthesis. ^1^
*p*-values of risk factors were determined with the logistic regression analysis. ^2^ Multivariate logistic regression (forward conditional method) was used to investigate the independent risk factors with the default probability of 0.05 for entry and 0.10 for removal. ^3^ Average of the measurements by two radiologists were involved in the statistical analysis.

## Data Availability

The anonymous data presented in this study are available on request from the corresponding author. The data are not publicly available due to institutional regulations.

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
