# Peer review of "C-Arm Cone-Beam CT Virtual Navigation versus Conventional CT Guidance in the Transthoracic Lung Biopsy: A Case-Control Study"

_diagnostics, 2022, doi:10.3390/diagnostics12010115_

Round 1

Reviewer 1 Report

Dear authors,

1, Similar research has been found (PMID:26310371). This makes the topic isn't of great enough significance. Could you please go through other similar research and introduce what's new in your manuscript?

2, Data presentation is nice. I think moderate English Editing is required.

Author Response

We appreciated your comment and offering of this similar article. First, we’d like to explain the differences between ours and the previous study that you mentioned [PMID:26310371]. In this previous study, the authors only explored the diagnostic accuracies between conventional CT- and CBCT-guided biopsies, which revealed a comparable result similar to our findings. However, in our study, we comprehensively explored the potential risk factors of biopsy-related complications, including different CT guidance, prior emphysema history, patient position during the biopsy, lesion size, needle repositioning, lesion-skin distance along the needle path (from lesion border to skin), lesion-pleural distance along the needle path (from lesion border to pleura), needle-pleural angle, and the number of obtained samples between both biopsy techniques. To our knowledge, no previous studies explored this issue as comprehensively as we did. So, we think this is an essential supplement to previous research. As a result, we found only one factor - needle repositioning - significantly increases the risk of pneumothorax and pulmonary hemorrhage/hemoptysis. Besides, our study involved a larger scale of participants than the previous study. In the modification, we cited this reference and compared it with our study.

Second, we modified our English writing thoroughly in the modification to make our manuscript more readable and natural. 

Reviewer 2 Report

The manuscript is well done 

Author Response

We appreciate your reviewing of this manuscript and are very happy to obtain your approval on our study. 

Reviewer 3 Report

The article titled: C-arm cone-beam CT virtual navigation versus conventional CT guidance in the transthoracic lung biopsy: a case control study submitted to Diagnostics (SI: Advances in the Diagnosis of Lung Nodules), presents very interesting, novel and relevant study. Authors proposed C-arm cone-beam computed tomography (CBCT) virtual navigation-guided lung biopsy as an alternative to conventional CT-guided lung biopsy. Authors retrospectively reviewed 217 patients undergoing CT or CBCT navigated lung biopsy. Nevertheless article is very interesting does very poor conclusions and results description.  Please work more about those two subdivisions especially conclusions. 

Author Response

We appreciate your comments on our study. In the modification, we thoroughly modified our Results and Conclusion parts following your suggestions. Especially, we revised our previous poor conclusion to make a more evident standpoint as: “This retrospective case-control study demonstrates that both the conventional CT- and C-arm CBCT virtual navigation-guided lung biopsies are safe and accurate. However, the C-arm CBCT virtual navigation facilitates the needle positioning and reduces the radiation dose, compared to conventional CT guidance. Since needle repositioning significantly increased the incidence of biopsy-related complications, as revealed in our study, C-arm CBCT virtual navigation can be more suggested in transthoracic lung biopsy to reduce the necessity of needle repositioning and the subsequent complications.”